# Resurgence of malaria in Uganda despite sustained indoor residual spraying and repeated long lasting insecticidal net distributions

Adrienne Epstein[1,2]*, Catherine Maiteki-Sebuguzi[3], Jane F. Namuganga[4], Joaniter I. Nankabirwa[4,5], Samuel Gonahasa[4], Jimmy Opigo[3], Sarah G. Staedke[6], Damian Rutazaana[3], Emmanuel Arinaitwe[4], Moses R. Kamya[4,5], Samir Bhatt[7,8], Isabel Rodríguez-Barraquer[9], Bryan Greenhouse[9], Martin J. Donnelly[1], Grant Dorsey[9]

1 Department of Vector Biology, Liverpool School of Tropical Medicine, Liverpool, United Kingdom, 2 Department of Epidemiology and Biostatistics, University of California San Francisco, San Francisco, California, United States of America, 3 National Malaria Control Division, Ministry of Health, Kampala, Uganda, 4 Infectious Diseases Research Collaboration, Kampala, Uganda, 5 College of Health Sciences, Makerere University, Kampala, Uganda, 6 London School of Hygiene and Tropical Medicine, London, United Kingdom, 7 Department of Infectious Disease Epidemiology, Imperial College, St Mary's Hospital, London, United Kingdom, 8 Department of Public Health, Section of Epidemiology, University of Copenhagen, Copenhagen, Denmark, 9 Department of Medicine, University of California San Francisco, San Francisco, California, United States of America

* Adrienne.Epstein@lstmed.ac.uk

**Data Availability Statement:** The datasets generated during and/or analysed during the

## Abstract

Five years of sustained indoor residual spraying (IRS) of insecticide from 2014 to 2019, first using a carbamate followed by an organophosphate, was associated with a marked reduction in the incidence of malaria in five districts of Uganda. We assessed changes in malaria incidence over an additional 21 months, corresponding to a change in IRS formulations using clothianidin with and without deltamethrin. Using enhanced health facility surveillance data, our objectives were to 1) estimate the impact of IRS on monthly malaria case counts at five surveillance sites over a 6.75 year period, and 2) compare monthly case counts at five facilities receiving IRS to ten facilities in neighboring districts not receiving IRS. For both objectives, we specified mixed effects negative binomial regression models with random intercepts for surveillance site adjusting for rainfall, season, care-seeking, and malaria diagnostic. Following the implementation of IRS, cases were 84% lower in years 4–5 (adjusted incidence rate ratio [aIRR] = 0.16, 95% CI 0.12–0.22), 43% lower in year 6 (aIRR = 0.57, 95% CI 0.44–0.74), and 39% higher in the first 9 months of year 7 (aIRR = 1.39, 95% CI 0.97–1.97) compared to pre-IRS levels. Cases were 67% lower in IRS sites than non-IRS sites in year 6 (aIRR = 0.33, 95% CI 0.17–0.63) but 38% higher in the first 9 months of year 7 (aIRR = 1.38, 95% CI 0.90–2.11). We observed a resurgence in malaria to pre-IRS levels despite sustained IRS. The timing of this resurgence corresponded to a change of active ingredient. Further research is needed to determine causality.

current study are available in GitHub at the URL https://github.com/aeepstein/resurgence-irs-uganda.

**Funding:** This work was supported by the National Institutes of Health as part of the International Centers of Excellence in Malaria Research (ICMER) program (U19AI089674, GD). Funding for this work was also supported by the National Institute of Allergy and Infectious Diseases (F31AI150029, AE). The funders had no role in study design, data collection and analysis, decision to publish, or preparation of the manuscript.

**Competing interests:** The authors have declared that no competing interests exist.

## Introduction

Major gains have been made in reducing the burden of malaria in sub-Saharan Africa over the past two decades, resulting in a 44% decline in malaria deaths between 2000 and 2019 [1,2]. However, progress has slowed in recent years, particularly in highest burden countries [1]. A majority of the decline in malaria burden has been attributed to vector control interventions, including long-lasting insecticidal nets (LLINs) and indoor residual spraying of insecticide (IRS). Scale-up of LLINs coverage has occurred rapidly, with the proportion of households in sub-Saharan Africa with at least one LLIN increasing from 5% in 2000 to 68% in 2019 [1]. Conversely, the percent of at-risk populations covered by IRS has been much lower and even declined from 5% in 2010 to 2% in 2019 [1]. Challenges to the scale-up of IRS coverage include high cost, complex implementation logistics, and community acceptance [3]. Furthermore, evidence from controlled trials is mixed on the added benefit of implementing IRS in communities where LLIN coverage is high [4]. In addition, there are limited data on the impact of multiple rounds of IRS sustained over an extended period of time and the relative impact of switching to different formulations of IRS [5,6]. This information is critical for maximizing the effectiveness of IRS, particularly as new products are deployed.

Uganda is illustrative of high burden countries where progress in reducing malaria burden has slowed in recent years [1]. The Ugandan Ministry of Health has made a strong commitment to ensuring high LLIN coverage, delivering LLINs through 3 universal coverage campaigns (UCC) taking place in 2013–2014, 2017–2018, and 2020–21. These campaigns have been successful: for example, in 2018–2019, 83% of households reported owning at least one LLIN, the highest coverage globally [7]. IRS was reintroduced into Uganda in 2006 for the first time since the 1960s, however, geographic coverage of IRS to date has been much lower than LLINs. From 2007 to 2014, IRS was implemented in 10 districts in Northern Uganda initially using pyrethroid insecticides and later switching to a carbamate [8]. In 2014, the IRS campaign was moved to 14 historically high burden districts in North-Eastern Uganda in 2014. The IRS campaign in the northeast initially deployed a carbamate insecticide (2014–2016) before changing to an organophosphate for the third, fourth, and fifth years (2017–2019). In this part of the country, IRS has been considered highly successful: in the first 3 years of sustained IRS campaigns (2014–2017), malaria cases at sentinel surveillance sites in 5 of these districts fell by 50%, and in the fourth and fifth years (2018–2019), by 85% compared to pre-IRS levels [6].

In this study, we use data from a network of health facility-based malaria surveillance sites to examine the impact of sustained IRS in the 5 districts mentioned above. This study had two objectives. First (Objective 1), to estimate the impact of IRS on malaria burden in 5 districts relative to a baseline period before IRS was initiated after extending our prior analysis [6] to the 6th and 7th years of sustained IRS (January 2020 through October 2021). This extended evaluation period coincided with a shift in insecticide formulations from the organophosphate pirimiphos-methyl (Actellic 300CS) to products containing the active ingredient clothianidin alone (SumiShield) or combined with deltamethrin (Fludora Fusion WP-SB), the beginning of the COVID-19 pandemic, and the third LLIN UCC (2020–2021). Second (Objective 2), we compared temporal changes in malaria burden during the 6th and 7th years of sustained IRS (January 2020 through October 2021) at the 5 sites receiving sustained IRS to 10 sites in neighboring districts that have not received IRS since 2017.

## Materials and methods

### Study sites and vector control interventions

This study utilized enhanced health facility surveillance data from 15 sites in Uganda. Five of these sites are among the 14 districts that have received repeated rounds of IRS since

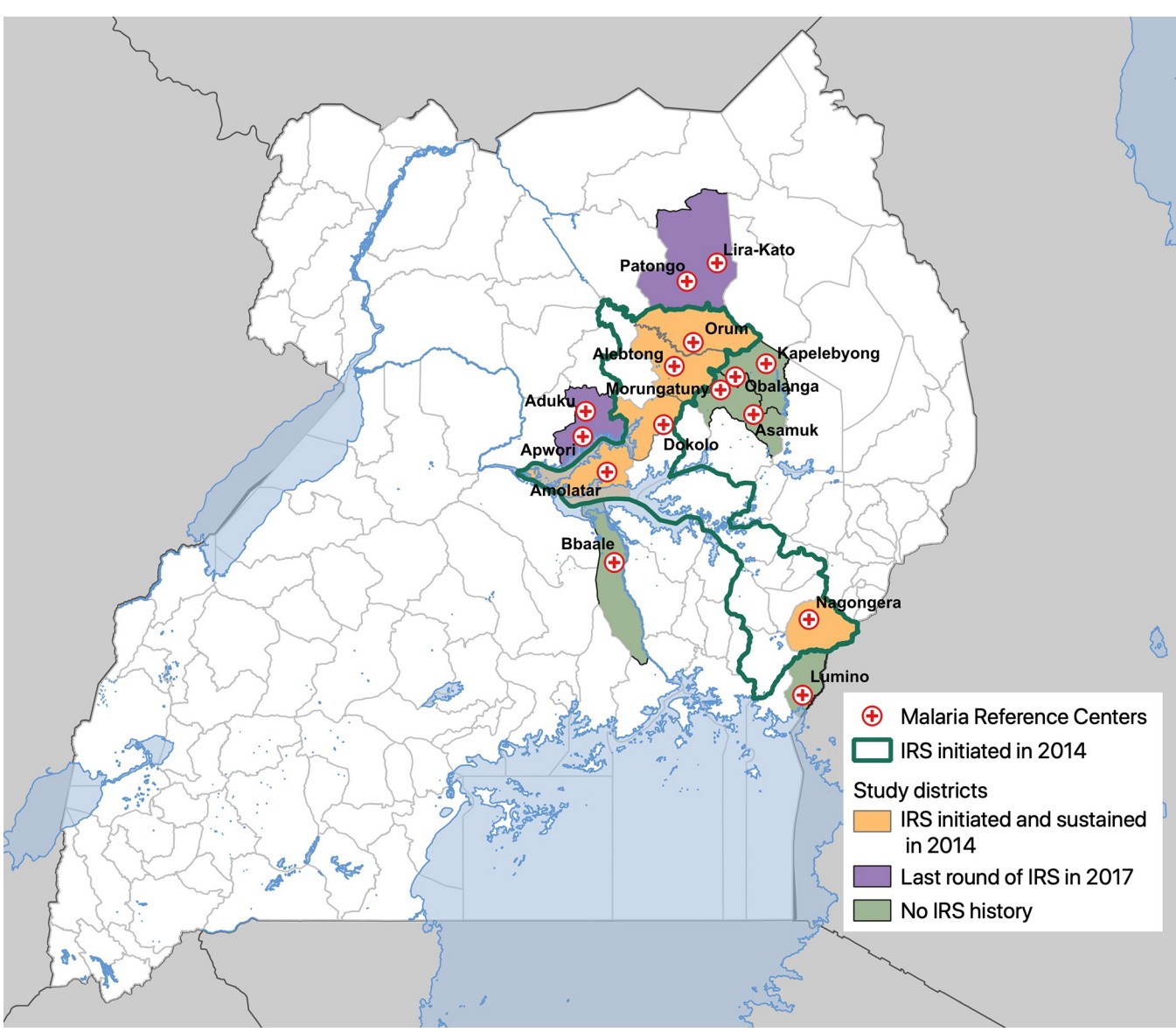

**Fig 1. Map of Uganda showing study sites.**

December 2014 (the remaining 9 districts did not have surveillance sites). Ten sites are in neighboring districts that have never received IRS (6 sites) or where IRS was last implemented in 2017 (4 sites, Fig 1). All sites included in the analysis are part of a surveillance network called the UMSP. National LLIN UCCs were conducted in 2013–2014, 2017–2018, and 2020–2021, where LLINs were distributed free-of-charge by the Uganda Ministry of Health targeting one LLIN for every two household residents throughout the entire country (following WHO recommendations [9]). During the most recent campaign (2020–2021), the Ministry of Health distributed conventional LLINs containing the pyrethroid deltamethrin (PermanNet 2.0) in addition to two types of "next generation" LLINs due to evidence of pyrethroid resistance: one containing deltamethrin and piperonyl butoxide (PBO; PermaNet 3.0) and one containing alpha-cypermethrin and pyriproxyfen (Royal Guard). Among the surveillance sites included in this study, the 5 sites in IRS districts were in sub-counties that received conventional

pyrethroid nets and the 10 sites in neighboring districts were in sub-counties that received "next generation" nets (5 received PermaNet 3.0 nets and 5 received Royal Guard nets).

In the 5 study districts where IRS was first implemented in late 2014, the insecticide formulation initially consisted of a carbamate (bendiocarb) with rounds repeated approximately every 6 months until 2016 when the active ingredient was changed to the organophosphate pirimiphos-methyl (Actellic 300CS) administered annually until 2019. In 2019, one study district (Dokolo) received a single round of IRS with SumiShield 50WG, a new IRS product containing clothianidin. In 2020, all IRS districts began receiving IRS with Fludora Fusion WP-SB, a new IRS insecticide containing a mixture of clothianidin and deltamethrin. Changes of IRS insecticides were made in accordance with the Ugandan Ministry of Health's insecticide resistance management plan which requires changing IRS formulations every three years to preempt the development of resistance [10,11]. In two study districts (Otuke and Alebtong), IRS was discontinued in 2021 due to lack of funding. For a complete timeline of all IRS campaigns, including insecticides and coverage, see S1 Table.

## Health-facility based surveillance

A network of enhanced malaria surveillance sites embedded in public health facilities was established by UMSP in 2006, as previously described [12]. In brief, UMSP currently operates Malaria Reference Centers (MRCs) at 70 level III/IV public health facilities across Uganda. Level III and IV health facilities are parish and sub-county level facilities, respectively, and provide diagnostic testing and treatment free of charge to populations of 20,000 to 100,000 people per facility. At MRCs, individual-level data are collected for all outpatients using outpatient registers (HMIS 002) and entered monthly into a database by on-site data entry officers. Information collected includes sociodemographic data (age, sex, and village of residence), whether malaria was suspected, results of laboratory testing for malaria (rapid diagnostic test [RDT] or microscopy), diagnoses (laboratory-confirmed and clinical), and treatments prescribed. UMSP places emphasis on high quality data, ensuring minimal missingness on key variables including age and place of residence, and providing training and materials to maximize diagnostic testing among patients suspected of having malaria. For Objective 1 (to evaluate the impact of IRS on malaria burden in 5 districts over 6.75 of sustained use), we utilized data from 5 MRCs from districts with where IRS was implemented in 2014. These MRC sites were selected because they had been active UMSP sites for at least 6 months prior to the initiation of IRS. To compare changes in malaria burden from January 2020 through October 2021 (years 6 & 7 following implementation of IRS) at sites receiving sustained IRS to sites in neighboring districts that have not received IRS recently, we used data from 10 additional MRC sites. These sites were selected because they were in districts that neighbor the 5 IRS sites and have been active UMSP sites since at least January 2020. Four of these sites were in 2 districts that received IRS in the past; however, IRS was last implemented in 2017. The impact of these previous IRS campaigns was no longer evident during the study period [6]. We opted to compare malaria burden in IRS versus non-IRS sites only during years 6 and 7 of IRS implementation given the availability of data in non-IRS sites.

## Measures

For Objective 1, the exposure was specified as an indicator variable representing each month since IRS was initiated. In separate models, an indicator variable, representing months 1–36 (years 1, 2, and 3), 37–60 (years 4 and 5), 61–72 (year 6), and 73–81 (the first 9 months of year 7) of sustained IRS was included as the primary exposure variable. Years 6 and 7 were presented separately as they represent data unpublished in our previous analysis (February 2020

through October 2021); we therefore believed temporal granularity was warranted. The baseline period was defined as the 12 months before IRS was implemented in 2014; if a site had less than 12 months of baseline data available, we included the maximum amount of time available.

For Objective 2, the primary exposure was specified as a binary variable representing whether a site was an IRS site or a non-IRS site.

For both objectives, the primary outcome was the monthly count of laboratory-confirmed malaria cases at each site. To correct for monthly testing rates, we adjusted this count by multiplying the number of individuals with suspected malaria but not tested each month by the test positivity rate (TPR, the number who tested positive divided by the total number tested) in that month. We then added the result to the number of laboratory-confirmed positive cases in that month. As a sensitivity analysis, we re-specified the models including only laboratory-confirmed case counts as the outcome.

We adjusted for time-varying variables that impact malaria burden and malaria case detection at the health facility surveillance site. This includes monthly precipitation [13] which was modeled non-linearly using restricted cubic splines. Lags of 0, 1, 2, and 3 months were considered for precipitation; the appropriate lag was selected by running univariable regressions with each lag and selecting that which demonstrated the lowest Akaike's information criterion (AIC). We also included indicator variables for month of the year (to adjust for season), the proportion of tests that were RDT (vs. microscopy) in that month, and the number of individuals who attended the health facility surveillance site but were not suspected of having malaria in that month (to adjust for potential changes in care-seeking behaviors, particularly during the COVID-19 lockdown in Uganda).

## Statistical analysis

For Objective 1, we specified mixed effects negative binomial regression models with random intercepts for site [6]. Coefficients for the exposure variable were exponentiated to represent the incidence rate ratio (IRR) comparing the incidence of malaria in the month of interest relative to the baseline pre-IRS period. These models test the null hypothesis of no difference of changes in IRS burden after the initiation of IRS compared to before the initiation of IRS, adjusting for seasonal effects and time-varying changes in diagnostic testing and care seeking.

For Objective 2, models were specified as mixed effects negative binomial regression models with random intercepts for site. The binary indicator variable representing whether a site was an IRS site or a non-IRS site was interacted with an indicator variable representing time (an indicator representing month/year and, in a separate model, a categorical variable representing January 2020-December 2020 and January 2021-October 2021). Coefficients for the exposure variable combined with the interaction term were exponentiated to represent the incidence rate ratio (IRR) comparing the burden of malaria IRS sites versus non-IRS sites over a given period of time. These models test the null hypothesis of no difference between malaria burden at IRS sites compared to non-IRS sites, adjusting for seasonal effects and time-varying changes in diagnostic testing and care seeking.

## Results

### Study objective 1

Across the 5 sites receiving sustained IRS, a total of 769,561 outpatient visits were recorded from the baseline period covering up to 12 months before IRS started through October 2021 (Table 1). During the baseline period, average monthly cases adjusted for testing ranged from 278–657 and TPR ranged from 25.2%-67.0%. By years 4 and 5 (months 37 to 60) of sustained

**Table 1. Summary statistics from health-facility based surveillance sites for study objective 1.**

| MRC (District) | Time period | Total outpatient visits, n | Suspected malaria cases, n (% of outpatient visits) | Tested for malaria, n (% of suspected malaria cases) | RDT performed (versus microscopy), n (% of tested) | Confirmed malaria cases, n (% of tested [TPR]) | Confirmed malaria cases adjusted for testing | Mean monthly malaria cases adjusted for testing |
|---|---|---|---|---|---|---|---|---|
| Nagongera HCIV (Tororo) | Baseline (12 months pre-IRS) | 20,828 | 13,251 (63.6) | 13,096 (98.8) | 760 (5.8) | 3,298 (25.2) | 3,337 | 278 |
| | Months 1–36 of IRS | 63,289 | 23,315 (38.4) | 24,084 (99.0) | 9,615 (40.0) | 4,004 (16.6) | 4,042 | 112 |
| | Months 37–60 of IRS | 35,745 | 12,616 (35.2) | 12,608 (99.9) | 3,798 (30.1) | 1,067 (8.5) | 1,067 | 30 |
| | Months 61–81 of IRS | 38,462 | 16,237 (42.2) | 16,233 (99.9) | 12,219 (75.3) | 4,386 (27.0) | 4,386 | 209 |
| Amolatar HCIV (Amolatar) | Baseline (12 months pre-IRS) | 19,552 | 8,547 (43.7) | 6,512 (76.2) | 5,923 (91.0) | 3,701 (56.8) | 4,845 | 404 |
| | Months 1–36 of IRS | 55,570 | 18,118 (32.6) | 15,082 (83.2) | 13,440 (89.1) | 3,924 (26.0) | 4,956 | 138 |
| | Months 37–60 of IRS | 35,231 | 7,038 (20.0) | 7,034 (99.9) | 6,279 (89.3) | 908 (12.9) | 1,088 | 30 |
| | Months 61–81 of IRS | 34,512 | 13,211 (38.3) | 13,180 (99.8) | 11,176 (84.8) | 6,974 (52.9) | 6,992 | 332 |
| Dokolo HCIV (Dokolo) | Baseline (12 months pre-IRS) | 25,570 | 12,854 (50.3) | 8,875 (69.0) | 8,212 (92.5) | 5,211 (58.7) | 7,889 | 657 |
| | Months 1–36 of IRS | 78,969 | 30,846 (39.1) | 29,476 (95.6) | 27006 (91.6) | 7,734 (26.2) | 8,266 | 230 |
| | Months 37–60 of IRS | 52,550 | 16,361 (31.1) | 16,273 (99.5) | 15,997 (98.3) | 2,524 (15.6) | 3,243 | 90 |
| | Months 61–81 of IRS | 54,489 | 23,142 (42.5) | 23,084 (99.8) | 20,671 (89.5) | 10,887 (47.2) | 10,913 | 520 |
| Orum HCIV (Otuke) | Baseline (11 months pre-IRS) | 16,120 | 9,324 (57.8) | 8,929 (95.8) | 3,990 (44.7) | 5,974 (66.9) | 6,236 | 566 |
| | Months 1–36 of IRS | 42632 | 26,642 (62.5) | 25,583 (96.0) | 11008 (43.0) | 13,619 (53.2) | 14,207 | 394 |
| | Months 37–60 of IRS | 23,424 | 11,064 (47.2) | 11,064 (100.0) | 8,717 (78.9) | 2,911 (26.3) | 3,072 | 85 |
| | Months 61–81 of IRS | 21,092 | 13,875 (65.8) | 13,875 (100.0) | 8,287 (59.7) | 6,979 (50.3) | 6,979 | 332 |
| Alebtong HCIV (Alebtong) | Baseline (8 months pre-IRS) | 15,359 | 6,694 (43.6) | 4,789 (71.5) | 4,620 (96.5) | 3,209 (67.0) | 4,317 | 540 |
| | Months 1–36 of IRS | 62,161 | 30,226 (48.6) | 25,863 (85.6) | 22,373 (86.5) | 10,452 (40.4) | 12,251 | 340 |
| | Months 37–60 of IRS | 33,201 | 11,091 (33.4) | 10,810 (97.5) | 10,399 (96.2) | 1,638 (15.1) | 1,745 | 48 |
| | Months 61–81 of IRS | 39,711 | 22,318 (56.2) | 22,318 (100.0) | 19,121 (85.7) | 11,155 (50.0) | 11,155 | 531 |

IRS these metrics had decreased to 30–90 and 8.5%-26.3%, respectively. However, in year 6 and the first 9 months of year 7 (months 61 to 81) of sustained IRS, average monthly cases adjusted for testing increased to a range of 209–431 and TPR ranged from 27.0%-52.9%. Fig 2 shows plots of laboratory-confirmed malaria cases and vector control interventions over time across the 5 sites. Each of these sites demonstrate similar patterns of a decline in malaria cases

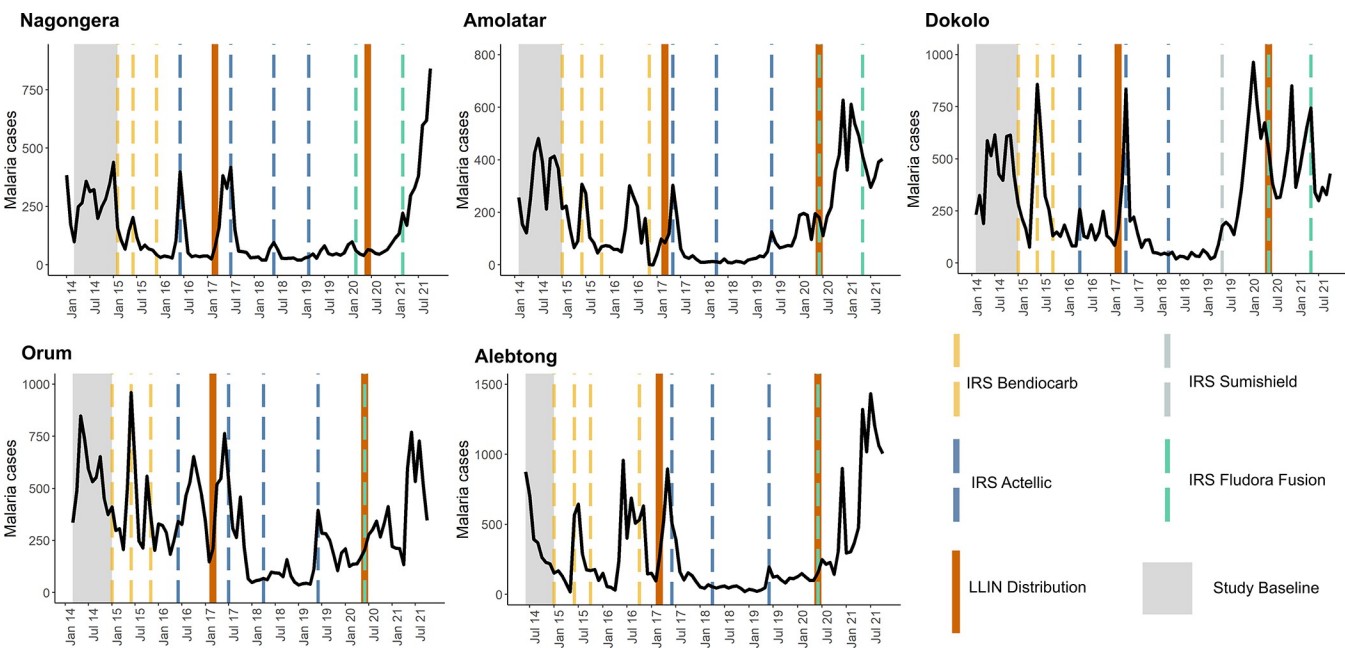

**Fig 2. Malaria case counts and vector control interventions over time at 5 IRS sites.** The study baseline period (pre-IRS) is indicated in grey.

after the initiation of IRS with seasonal peaks during the first three years of IRS (through 2017), a substantial decline in burden in years four and five of sustained IRS (through 2019), and an increase in burden in years 6 and 7 of sustained IRS (2020–2021) equal to or higher than the burden in the baseline period before IRS was implemented.

Monthly adjusted IRRs and 95% confidence intervals (CI) for the 5 sites combined are presented in Fig 3 and in S2 Table. These results show there was an initial 52% (95% CI 36% to 64%) reduction in malaria burden in months 1 through 36 of sustained IRS (adjusted IRR = 0.48, 95% CI 0.36–0.64), followed by a continued improvement to an 84% (95% CI 78% to 88%) reduction in burden in months 37 through 60 of sustained IRS (adjusted IRR = 0.16, 95% CI 0.12–0.22). In months 61 through 72, however, malaria burden increased and was only 43% (95% CI 26% to 56%) lower than the baseline period before IRS was implemented (adjusted IRR = 0.57, 95% CI 0.44–0.74). During months 73 through 81 after the initiation of IRS, there was a trend towards a higher malaria burden than the pre-IRS period (adjusted IRR = 1.39, 95% CI 0.97–1.97). These results were consistent when including only laboratory-confirmed cases unadjusted for testing rate as the model outcome (S1 Fig, S2 Table) and when repeating the analysis leaving out the 2 sites that halted IRS campaigns after 2020 (Orum and Alebtong) (S2 Fig).

### Study objective 2

Across the 15 sites over a 22 month period (January 2020 through October 2021) included in the analysis for Objective 2, 583,344 outpatient visits were recorded (Table 2). From January 2020 through December 2020, average monthly cases at IRS sites ranged from 64–594 and TPR ranged from 10.9%-51.0%. From January 2021 through October 2021, these figures increased to 371–849 and 43.4%-60.1%, respectively. Average monthly cases increased in 4 of the 5 IRS sites comparing January 2020 through December 2020 to January 2021 through October 2021, with the exception of Dokolo (the first site to switch to IRS with a clothianidin-

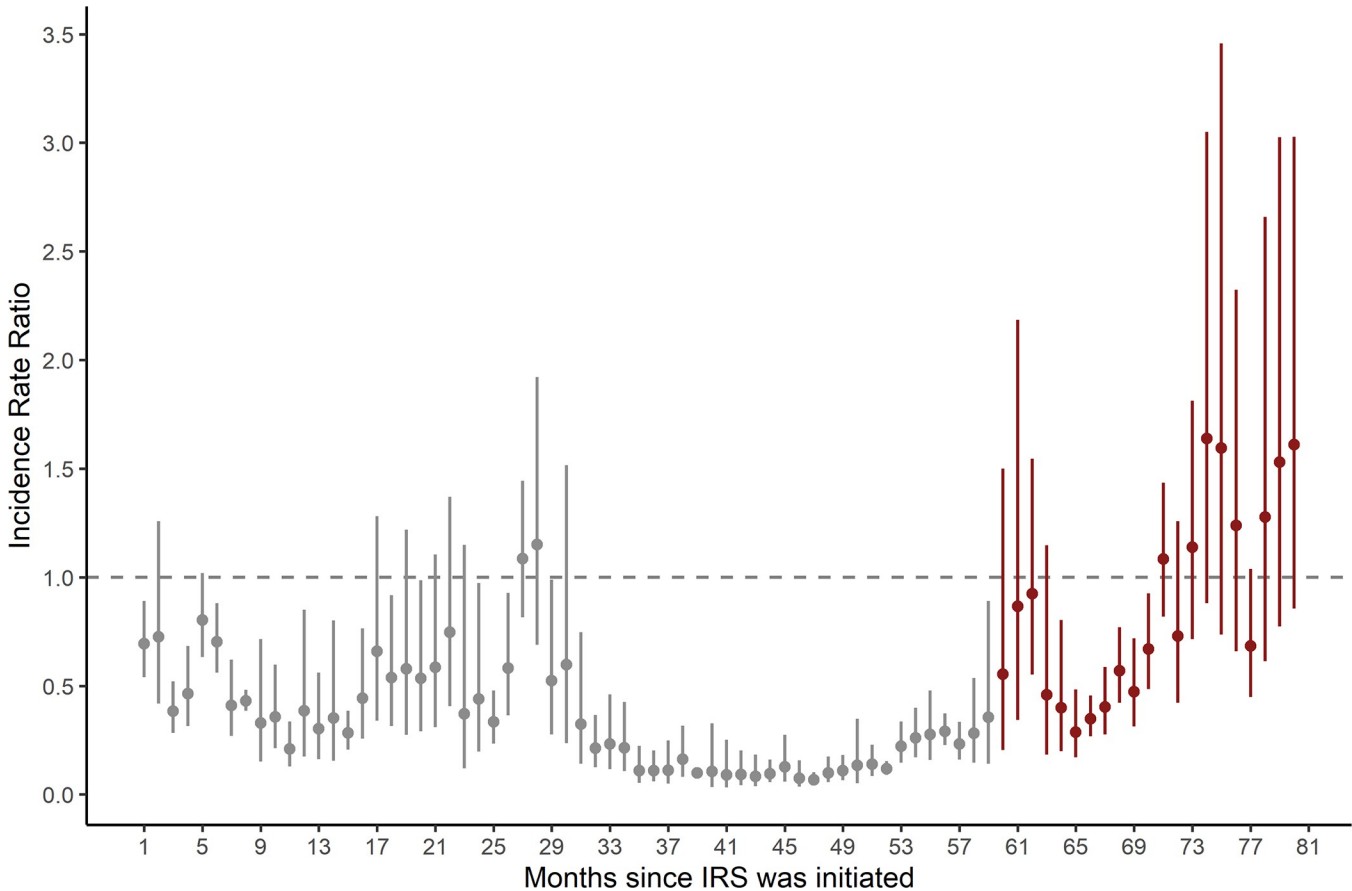

**Fig 3. Adjusted IRR from multilevel negative binomial model comparing the period after IRS was initiated to the period before IRS was initiated.**
Vertical bars represent the 95% CI around adjusted IRR. Effect estimates in grey are published previously.

based formulation in 2019), which already experienced a large increase in monthly malaria cases by January 2020.

In sites that had not received IRS recently, average monthly cases from January 2020 through December 2020 ranged from 518–1,068 and TPR from 47.5%-71.1%. These figures decreased to 251–716 and 32.5%-50.4%, respectively, from January 2021 through October 2021. Fig 4 shows plots of laboratory-confirmed malaria cases and the timing of the recent LLIN distribution across the 10 non-IRS sites from January 2020 through October 2021. These figures indicate a substantial increase in cases the first year of observation for some sites, but a general downward trend in cases across the 10 sites over the 22 month study period for Objective 2.

Fig 5 shows the mean number of cases across the 5 IRS sites and the 10 sites that have not received IRS recently from January 2020 to October 2021 (corresponding to months 60 through 81 of IRS for the 5 sites where IRS was initiated and sustained). This figure shows an increase in cases at IRS sites, particularly in the latter half of the observation window, while cases at non-IRS sites trended downward over the same time period. Fig 5 also shows the adjusted IRR comparing cases at IRS sites to sites that have not received IRS recently. Coefficients and 95% confidence intervals are also presented in S3 Table. These findings indicate that from January 2020 to November 2020, cases were significantly lower in IRS sites compared to non-IRS sites. From December 2020 onward, the 95% CI for the IRR crossed 1, and

**Table 2. Summary statistics from health-facility based surveillance sites for study objective 2.**

| MRC (District) | Time period | Total outpatient visits, n | Suspected malaria cases, n (% of outpatient visits) | Tested for malaria, n (% of suspected malaria cases) | RDT performed (versus microscopy), n (% of tested) | Confirmed malaria cases, n (% of tested [TPR]) | Confirmed malaria cases adjusted for testing | Mean monthly malaria cases adjusted for testing |
|---|---|---|---|---|---|---|---|---|
| | | | | **IRS SITES** | | | | |
| Nagongera HCIV (Tororo) | January-December 2020 | 20,079 | 7,033 (35.0) | 7,031 (100.0) | 5,706 (69.2) | 767 (10.9) | 767 | 64 |
| | January-October 2021 | 20,405 | 9,827 (48.2) | 9,825 (99.9) | 6,797 (83.3) | 3,706 (57.7) | 3,706 | 371 |
| Amolatar HCIV (Amolatar) | January-December 2020 | 18,995 | 6,360 (33.5) | 6,349 (99.8) | 5,198 (81.9) | 2,965 (46.7) | 2,969 | 247 |
| | January-October 2021 | 17,230 | 7,311 (42.2) | 7,289 (99.7) | 6,415 (88.0) | 4,198 (57.6) | 4,213 | 421 |
| Dokolo HCIV (Dokolo) | January-December 2020 | 32,042 | 13,984 (43.6) | 13,947 (99.7) | 12,842 (92.1) | 7,110 (51.0) | 7,130 | 594 |
| | January-October 2021 | 25,250 | 10,477 (41.6) | 10,446 (99.7) | 9,117 (87.2) | 4,519 (43.3) | 4,531 | 453 |
| Orum HCIV (Otuke) | January-December 2020 | 11,161 | 6,807 (61.0) | 6,807 (100.0) | 5,672 (83.3) | 2,925 (43.0) | 2,925 | 244 |
| | January-October 2021 | 10,955 | 7,632 (70.0) | 7,632 (100.0) | 3,126 (41.0) | 4,264 (55.9) | 4,264 | 426 |
| Alebtong HCIV (Alebtong) | January-December 2020 | 19,027 | 8,877 (46.7) | 8,877 (100.0) | 7,763 (87.5) | 2,797 (31.5) | 2,797 | 233 |
| | January-October 2021 | 21,861 | 13,953 (63.4) | 13,953 (100.0) | 11,851 (84.9) | 8,485 (60.8) | 8,485 | 849 |
| | | | | **NON-IRS SITES** | | | | |
| Aduku HCIV (Kwania) | January-December 2020 | 26,537 | 19,136 (72.1) | 19,133 (99.9) | 12,770 (66.7) | 12,818 (67.0) | 12,821 | 1,068 |
| | January-October 2021 | 22,196 | 14,220 (64.1) | 14,213 (100.0) | 7,308 (51.4) | 7,160 (50.4) | 7,164 | 716 |
| Patongo HCIII (Agago) | January-December 2020 | 21,138 | 18,076 (85.6) | 18,071 (99.9) | 17,417 (96.4) | 11,911 (65.9) | 11,914 | 993 |
| | January-October 2021 | 11,656 | 9,023 (77.4) | 9,023 (100.0) | 7,522 (83.4) | 4,009 (44.4) | 4,009 | 401 |
| Bbaale HCIV (Kayunga) | January-December 2020 | 25,382 | 15,691 (61.8) | 15,591 (99.4) | 11,856 (76.0) | 7,401 (47.5) | 7,450 | 621 |
| | January-October 2021 | 15,266 | 8,185 (53.6) | 8,154 (99.6) | 6,187 (75.9) | 2,502 (30.7) | 2,511 | 251 |
| Lumino HCIII (Busia) | January-December 2020 | 21,885 | 17,783 (81.2) | 17,783 (100.0) | 16,094 (90.5) | 10,409 (58.5) | 10,409 | 867 |
| | January-October 2021 | 14,354 | 11,018 (76.8) | 11,018 (100.0) | 8,542 (77.5) | 5,435 (49.3) | 5,435 | 544 |

*(Continued)*

**Table 2.** (Continued)

| MRC (District) | Time period | Total outpatient visits, n | Suspected malaria cases, n (% of outpatient visits) | Tested for malaria, n (% of suspected malaria cases) | RDT performed (versus microscopy), n (% of tested) | Confirmed malaria cases, n (% of tested [TPR]) | Confirmed malaria cases adjusted for testing | Mean monthly malaria cases adjusted for testing |
|---|---|---|---|---|---|---|---|---|
| Apwori HCIII (Kwania) | January–December 2020 | 14,759 | 13,783 (93.4) | 13,781 (99.9) | 12,189 (88.5) | 9,164 (66.5) | 9,166 | 764 |
| | January–October 2021 | 9,506 | 8,058 (84.8) | 8,058 (100.0) | 8,014 (99.5) | 2,826 (35.1) | 2,826 | 283 |
| Lira-Kato HCIII (Agago) | January–December 2020 | 20,059 | 17,825 (88.9) | 17,814 (99.9) | 17,801 (99.9) | 12,670 (71.1) | 12,679 | 1057 |
| | January–October 2021 | 12,064 | 9,715 (80.5) | 9,714 (100.0) | 6,871 (70.7) | 5,313 (54.7) | 5,313 | 531 |
| Morungatuny HCIII (Amuria) | January–December 2020 | 15,053 | 13,964 (92.7) | 13,964 (100.0) | 13,940 (99.8) | 7,698 (55.1) | 7,698 | 642 |
| | January–October 2021 | 10,320 | 9,759 (94.5) | 9,759 (100.0) | 9,758 (100.0) | 3,168 (32.5) | 3,168 | 317 |
| Asamuk HCIII (Amuria) | January–December 2020 | 20,504 | 18,065 (88.1) | 18,045 (99.9) | 17,633 (97.7) | 10,728 (59.5) | 10,742 | 895 |
| | January–October 2021 | 18,230 | 15,668 (85.9) | 15,668 (100.0) | 15,387 (98.2) | 6,325 (40.4) | 6,325 | 633 |
| Kapelebyong HCIV (Kapelebyong) | January–December 2020 | 16,975 | 10,720 (63.2) | 10,711 (99.9) | 10,652 (99.5) | 6,215 (58.0) | 6,219 | 518 |
| | January–October 2021 | 15,579 | 7,858 (50.4) | 7,858 (100.0) | 7,849 (99.9) | 2,596 (33.0) | 2,596 | 260 |
| Obalanga HCIII (Kapelebyong) | January–December 2020 | 17,438 | 15,855 (90.9) | 15,852 (99.9) | 12,833 (81.0) | 9,917 (62.6) | 9,919 | 827 |
| | January–October 2021 | 12,438 | 11,397 (91.6) | 11,397 (100.0) | 9,834 (86.3) | 5,079 (41.6) | 5,079 | 508 |

from March 2021 onward the point estimate for the IRR crossed 1, indicating that cases at IRS sites were higher than non-IRS sites, although we could not rule out a null or negative association in these months. These results show that cases were 67% lower (95% CI 37% to 83%) in IRS sites than non-IRS sites from January 2020 through December 2020 (adjusted IRR = 0.33, 95% CI 0.17–0.63) and 38% (95% CI -10% to 211%) higher from January through October 2021 (adjusted IRR = 1.38, 95% CI 0.90–2.11). Results were consistent when including only laboratory-confirmed cases unadjusted for testing as the outcome (S3 Fig, S3 Table).

## Discussion

We utilized enhanced health facility surveillance data from 5 sites in North-Eastern Uganda that have undergone sustained IRS since 2014 to evaluate the impact of repeated IRS campaigns in their 6th and 7th years. Our findings point to a resurgence in the burden of malaria at

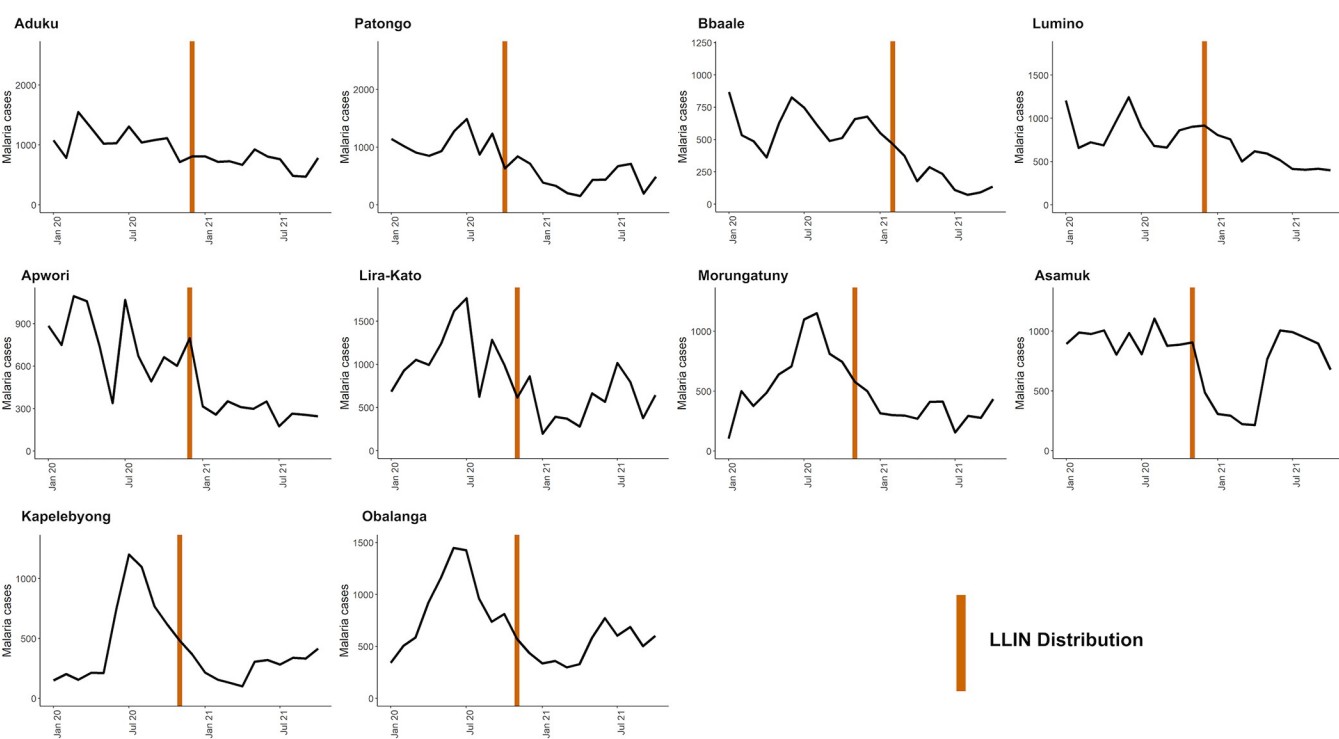

**Fig 4. Malaria case counts and vector control interventions over time at 10 sites that have not received IRS recently.**

these 5 surveillance sites, despite sustained IRS and repeated LLIN UCCs conducted every 3 years. In the final 9 months of observation (months 61–81 of sustained IRS), malaria burden reached similar levels, and in some instances higher levels, than the period before IRS was initiated in late 2014. We did not observe corresponding increases in burden at 10 surveillance sites in neighboring non-IRS districts that, unlike IRS districts, received "next generation" LLINs over the study period. These findings suggest that in the setting of universal coverage with conventional pyrethroid-only LLINs, the marked benefit of adding sustained IRS over the

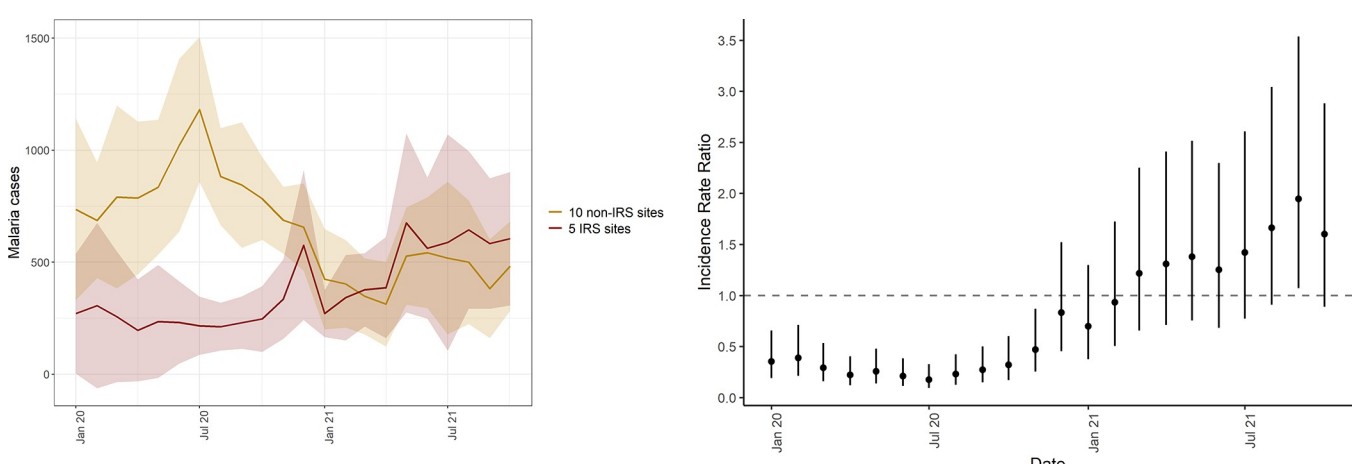

**Fig 5. Mean case counts in 5 IRS sites and 10 non-IRS sites from January 2020 through July 2021 (left) and adjusted IRR comparing IRS sites to non-IRS sites over the study period.** Shaded areas represent standard deviations. Vertical bars represent the 95% CI around adjusted IRR.

first five years had been lost over the subsequent 18 months that coincided with a change in active ingredient and the start of the COVID-19 pandemic.

This study highlights the importance of high quality routine surveillance to monitor the impact of population level malaria control interventions using clinically relevant indicators such as symptomatic cases diagnosed at health facilities [14–17]. The assessment of vector control interventions often focuses on entomologic outcomes such as mosquito mortality in controlled settings. This is understandable, given that measuring the impact of control interventions on clinical burden in "real world" settings is difficult. Nevertheless, the findings from this analysis underscore the potential feasibility of and need for robust epidemiologic surveillance systems to document the impact of vector control interventions on population health as they are applied. Unlike with standard HMIS data, UMSP surveillance sites have remarkably high reporting, testing, and proper treatment rates, with extremely low missingness of key variables; these data therefore lend themselves well to accurately assessing changes in malaria incidence over time.

Routine surveillance in areas undergoing IRS is particularly necessary because published studies from controlled trials on the added value of IRS in areas with high LLIN coverage has produced mixed results. A recent Cochrane review reported that adding IRS using a "pyrethroid-like" insecticide to LLINs did not provide any benefits, while adding IRS with a "non-pyrethroid-like" insecticide produced mixed results [4]. It is of note that none of the trials that evaluated the impact of adding IRS with a "non-pyrethroid-like" insecticide assessed outcomes beyond two years. More recently, observational studies evaluating the effectiveness of pirimiphos-methyl (Actellic 300CS) in "real world" settings have documented impressive impacts of IRS in Mali [18,19], Ghana [20], Zambia [5], Kenya [21], and Uganda [6]. This analysis contributes to observational data on the impacts of sustained IRS beyond 6 years, in the context of the COVID-19 pandemic and a switch to products containing the active ingredient clothianidin.

There are several potential factors that may be driving the increase in malaria burden in districts receiving IRS. First, the timing of the increase corresponds to a shift in active ingredient from pirimiphos-methyl to clothianidin-based formulations (primarily Fludora Fusion, a combination of clothianidin and deltamethrin). One reason clothianidin may be less effective than pirimiphos-methyl in this setting is that is slower-acting; if a proportion of mosquitoes exposed to clothianidin are surviving long enough to bite or lay eggs, this may lead to lower effectiveness of clothianidin compared to the fast-acting organophsopate pirimiphos-methyl. Of note, the one site (Dokolo) that received a single round of IRS with clothianidin-based SumiShield 50WG in 2019 (before the shift to Fludora Fusion WP-SB across all sites in 2020) experienced an increase in burden immediately following the round of SumiShield 50WG. Increases were not documented at other sites until the initiation of Fludora Fusion WP-SB in 2020. This is added evidence for the change in active ingredient contributing to the observed resurgence of malaria. To date, studies on the effectiveness of clothianidin alone (SumiShield) [22–26] and clothianidin with deltamethrin (Fludora Fusion WP-SB) [22,27–29] are limited to susceptibility studies focused on entomological outcomes, both in lab and in experimental huts. These studies document that clothianidin-based products succeed at killing mosquitoes, but none have evaluated clinically relevant outcomes in "real world" settings.

Another second potential driver that could explain the observed resurgence is that IRS districts received conventional pyrethroid LLINs in the 2020–2021 UCC campaign. This could be problematic given that wide-spread pyrethroid resistance has been described in Uganda [30–32] and may have increased since the 2017 UCC. Conversely, the 10 sites in neighboring districts where increases in burden were not observed received "next generation" nets designed for areas with wide-spread pyrethroid resistance. Therefore, the dual impact of IRS and LLINs

in these districts where pyrethroid resistance is high or increasing may have been lost between the two LLIN distributions. Importantly, there are no data on the impact of IRS with clothianidin in combination with LLINs (pyrethroid or next-generation). This is a key area for future research.

A third potential contributor to the observed resurgence of malaria at sites receiving IRS is the COVID-19 pandemic. An important concern has been the potential for delayed or inadequate implementation of vector control measures due to the pandemic [33,34]. In Uganda, the LLIN distribution campaign was delayed by 5 months but successfully distributed over 28 million nets, achieving over 90% coverage; in both IRS and non-IRS districts included in this analysis coverage was greater than 97%. Given that the delay in the UCC was country-wide, we do not believe an increase in burden resulting from delayed net distribution would be observed strictly in IRS sites as was the case in this study. The implementation of IRS campaigns did not appear impacted by the COVID-19 pandemic; annual campaigns in 2020 and 2021 were not delayed and coverage remained high (>90%, see S1 Table). We cannot rule out the potential for undocumented differences in implementation (for example, if spray operators spent less time in homes due to fear of acquiring COVID-19) that may have contributed to the observed resurgence. The pandemic may also have impacted patient behavior; for example, patients may have delayed seeking care for malarial illness which could have led to increased transmission. However, because sites that have not recently received IRS did not observe an increase in transmission, this explanation is less tenable. Furthermore, an analysis assessing the potential impact of the first year of the COVID-19 pandemic at Uganda Malaria Surveillance Program (UMSP) sites found no impact of the pandemic on malaria cases and non-malarial visits at health facilities [35].

Another potential explanation is a shift in mosquito species composition or mosquito behavior to outdoor biting, circumventing vector control interventions that target indoor biting mosquitoes [36]. This, however, would be unlikely to have occurred rapidly and simultaneously at all IRS sites at a pace that would explain the resurgence observed in this study and is not echoed in PMI Vectorlink entomological surveillance reports from 2019 and 2020, which indicate that IRS is not associated with increases in outdoor biting [37,38]. In addition, this would not explain the observation that the resurgence was observed only in the IRS sites and not in the non-IRS sites that only received LLINs, as a shift to outdoor biting would have a negative impact on both IRS and LLINs. Similarly, a shift in the predominant species from *Anopheles gambiae* and *Anopheles funestus* to *Anopheles arabiensis* may have led to a change from predominately indoor to outdoor biting [39]. However, recent data demonstrate that *Anopheles gambiae* and *Anopheles funestus* remain the primary vectors in Uganda [32], including in IRS districts (including Otuke and Tororo) according to 2020 entomological surveillance data [38].

This study is not without limitations. First, we used an observational study design, with measures of impact based on comparisons made before-and-after the implementation of IRS and comparisons of IRS and non-IRS districts. While cluster randomized controlled trials remain the gold standard design for estimating the impact of IRS, withholding IRS may be unethical, given what is known about its beneficial impacts, particularly in Uganda [6]. Similarly, comparisons of IRS sites and sites that have not recently received IRS are strictly descriptive, given that districts receiving IRS and not receiving IRS are not exchangeable. Second, while we hypothesize potential mechanisms that may explain the observed resurgence in malaria burden in IRS sites, we cannot rule potential secular trends or other unmeasured contributing factors. For example, while we attempt to control for seasonal effects and climate-related drivers of malaria burden over time, it is certainly possible that more recent months over the observation period correspond to an unusually high burden period in Uganda.

However, overall secular trends are an unlikely cause given the contemporaneous decline in malaria burden at nearby non-IRS sites. Third, the outcome for this analysis is limited to case counts of laboratory-confirmed malaria captured at health facilities. We do not have additional data on other metrics of transmission intensity, including entomologic measures, residual efficacy, nor on malaria mortality.

Despite these limitations, this analysis has important scientific and policy implications. First, additional research is needed on the driving factors contributing to the observed resurgence of malaria burden in IRS districts, including targeted entomology and residual efficacy. Data from PMI Vectorlink in 2020 in Tororo district suggest that Fludora Fusion effectiveness extends up to 8 months depending on the wall type, with lower effectiveness on mud walls [38]; however, more data, with both greater spatial and longitudinal variation, are needed. Furthermore, the Ministry of Health should be prepared to make timely changes in malaria control interventions based on on-going surveillance. For example, in IRS districts, policy makers may consider switching to the use of newer generation LLINs containing PBO which have been shown to be more effective than traditional pyrethroid treated LLINs in Uganda [40]. Future changes to the IRS active ingredient should take into consideration on-going surveillance, in line with Uganda's resistance management strategy. Consideration should also be made to key logistical factors including cost, procurement, and community acceptability. The unprecedented increase in malaria burden in areas where incidence had declined by 85% underscores the need to remain vigilant and responsive. Indeed, two of the five IRS districts included in this analysis stopped receiving IRS altogether in 2021, highlighting the challenge of maintaining gains in the face of inadequate resources and the need for rational exit strategies when IRS cannot be sustained. Finally, maintaining high quality, continuous surveillance systems to assess the impact of population level malaria control interventions remains essential in order to generate timely, actionable data.

## Supporting information

**S1 Fig. Adjusted IRR from multilevel negative binomial model comparing the period after IRS was initiated to the period before IRS was initiated with unadjusted case counts as model outcome.** Vertical bars represent the 95% CI around adjusted IRR. Effect estimates in grey are published previously.
(TIF)

**S2 Fig. Adjusted IRR from multilevel negative binomial model comparing the period after IRS was initiated to the period before IRS was initiated leaving out 2 sites that stopped IRS in 2021 (Orum and Alebtong).** Vertical bars represent the 95% CI around adjusted IRR. Effect estimates in grey are published previously.
(TIF)

**S3 Fig. Adjusted IRR comparing IRS sites to non-IRS sites over the study period with unadjusted case counts as model outcome.** Vertical bars represent the 95% CI around adjusted IRR.
(TIF)

**S1 Table. Timing and formulation of IRS campaigns.**
(XLSX)

**S2 Table. Unadjusted and adjusted incident rate ratios and 95% for study objective 1 (outcome adjusted and unadjusted for testing rates).**
(XLSX)

**S3 Table. Unadjusted and adjusted incident rate ratios and 95% for study objective 2 (outcome adjusted and unadjusted for testing rates).**
(XLSX)

## Acknowledgments

We would like to acknowledge the health workers at the health facilities that contributed data for this study. We would like to thank the Ugandan Ministry of Health National Malaria Control Division, and USAID–President's Malaria Initiative.

## Author Contributions

**Conceptualization:** Adrienne Epstein, Catherine Maiteki-Sebuguzi, Jane F. Namuganga, Grant Dorsey.

**Data curation:** Adrienne Epstein, Jane F. Namuganga, Grant Dorsey.

**Formal analysis:** Adrienne Epstein.

**Funding acquisition:** Sarah G. Staedke, Moses R. Kamya, Grant Dorsey.

**Investigation:** Catherine Maiteki-Sebuguzi, Sarah G. Staedke, Isabel Rodríguez-Barraquer, Grant Dorsey.

**Methodology:** Adrienne Epstein, Samir Bhatt, Isabel Rodríguez-Barraquer, Bryan Greenhouse.

**Project administration:** Jane F. Namuganga, Joaniter I. Nankabirwa, Samuel Gonahasa, Sarah G. Staedke, Emmanuel Arinaitwe.

**Software:** Adrienne Epstein.

**Supervision:** Joaniter I. Nankabirwa, Samuel Gonahasa, Sarah G. Staedke, Emmanuel Arinaitwe, Moses R. Kamya, Samir Bhatt, Isabel Rodríguez-Barraquer, Bryan Greenhouse, Martin J. Donnelly, Grant Dorsey.

**Validation:** Jimmy Opigo, Damian Rutazaana, Grant Dorsey.

**Visualization:** Adrienne Epstein.

**Writing – original draft:** Adrienne Epstein, Grant Dorsey.

**Writing – review & editing:** Adrienne Epstein, Catherine Maiteki-Sebuguzi, Jane F. Namuganga, Joaniter I. Nankabirwa, Samuel Gonahasa, Jimmy Opigo, Sarah G. Staedke, Damian Rutazaana, Emmanuel Arinaitwe, Moses R. Kamya, Samir Bhatt, Isabel Rodríguez-Barraquer, Bryan Greenhouse, Martin J. Donnelly, Grant Dorsey.

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
