## [Decision Letter · Decision Letter 0]

15 May 2022

PGPH-D-22-00463

Resurgence of malaria in Uganda despite sustained indoor residual spraying and repeated long lasting insecticidal net distributions

Dear Dr. Epstein,

Thank you for submitting your manuscript to PLOS Global Public Health. After careful consideration, we feel that it has merit but does not fully meet PLOS Global Public Health’s publication criteria as it currently stands. Therefore, we invite you to submit a revised version of the manuscript that addresses the points raised during the review process.

Please submit your revised manuscript by . If you will need more time than this to complete your revisions, please reply to this message or contact the journal office at globalpubhealth@plos.org. Please include the following items when submitting your revised manuscript:

We look forward to receiving your revised manuscript.

Kind regards,

Ruth Ashton, Ph.D.

Academic Editor

Journal Requirements:

1. Please amend your Financial Disclosure statement. If you did not receive any funding for this study, please simply state: “The authors received no specific funding for this work.”

2. Please update your Competing Interests statement. If you have no competing interests to declare, please state: “The authors have declared that no competing interests exist.”

3. We do not publish any copyright or trademark symbols that usually accompany proprietary names, eg (R), (C), or TM  (e.g. next to drug or reagent names). Please remove all instances of trademark/copyright symbols throughout the text, including Actellic 300CS®, SumiShield®, Fludora®, Permanet®, Royal Guard®, Royal Guard® on pages 4, 5, 6, 17, .

4. All figures and supporting information files will be published under the Creative Commons Attribution License (creativecommons.org/licenses/by/4.0/). Authors retain ownership of the copyright for their article and are responsible for third-party content used in the article. 

Figure 1: please (a) provide a direct link to the base layer of the map used and ensure this is also included in the figure legend; (b) provide a link to the terms of use / license information for the base layer. We cannot publish proprietary or copyrighted maps (e.g. Google Maps, Mapquest) and the terms of use for your map base layer must be compatible with our CC-BY 4.0 license. 

Please upload any written confirmation as an 'Other' file type. It must clarify that the copyright holder understands and agrees to the terms of the CC BY 4.0 license; general permission forms that do not specify permission to publish under the CC BY 4.0 will not be accepted. Note that uploading an email confirmation is acceptable.

Additional Editor Comments (if provided):

Reviewers' comments:

Reviewer's Responses to Questions

**Comments to the Author**

1. Does this manuscript meet PLOS Global Public Health’s publication criteria? Is the manuscript technically sound, and do the data support the conclusions? The manuscript must describe methodologically and ethically rigorous research with conclusions that are appropriately drawn based on the data presented.

Reviewer #1: Yes

Reviewer #2: Yes

2. Has the statistical analysis been performed appropriately and rigorously?

Reviewer #1: Yes

Reviewer #2: Yes

3. Have the authors made all data underlying the findings in their manuscript fully available (please refer to the Data Availability Statement at the start of the manuscript PDF file)?

Reviewer #1: Yes

Reviewer #2: Yes

4. Is the manuscript presented in an intelligible fashion and written in standard English?

Reviewer #1: Yes

Reviewer #2: Yes

5. Review Comments to the Author

Reviewer #1: Overall, I think this is an excellent manuscript that is very clearly written. It addresses two under-researched areas – the impact of IRS over multiple years and some evidence, albeit not strong, of the epidemiological impact of IRS with clothianidin-based products.

I have only minor issues to report.

Minor issues:

1. Line 65-69: The first introduction paragraph is about the scale-up of LLINs and IRS, which doesn’t fit well with the main themes of the paper. I think the introduction would be stronger if it focused instead on the current evidence of what we know/don’t know about the evidence of IRS implementation. I would build on the mixed benefit statement and fill in other details re: we don’t know much about the duration (months)of impact from one campaign, impact of withdrawing the intervention, relative impact of IRS products, or the long-term impact of the interventions, beyond a few studies. This paper addresses the impact of 7 years of sustained IRS, addressing current gaps in the literature. Some of this content is included in the discussion and could be pulled into the introduction.

2. Line 112: would be good practice to cite the WHO recommendation re: one ITN per 1 people.

3. Line 112: Would be helpful to name the specific pyrethroid chemical and brand for the conventional LLINs, as with the PBO and dual-AI LLINs.

4. Line 131: Changing IRS insecticides every three years is not technically in line with WHO recommendations, although I understand why national programs take this approach due to the lack of options for new products. Per the WHO Guidelines Annex 9: “The pragmatic approach is to rotate insecticides annually. Changing insecticides more than once a year (which could be the case in areas where two spray rounds are conducted each year) is not recommended, mainly because of procurement and other logistical challenges. Changing less frequently than once a year is also not recommended, as using an insecticide for longer makes it more likely that resistance will evolve to a frequency that is too high for rotations then to be effective in reducing it.”

5. Line 153: Could you explain why Study objective 2, with the non-IRS comparison sites only went back 2 years, creating a non-randomized post-only study design? Even though there is no randomization, I think it would be stronger if it looked at total trends in both IRS and non-IRS areas since the baseline period, but I understand if this is not possible because the data was not available.

6. Line 162: Would be nice to add a sentence about why the exposure variable was lumped into years 1-3, years 4-5, then separating out years 6 and year 7.

7. Line 276: The text says: “From December 2020 onward, the IRR crossed 1, indicating that cases at IRS sites were higher than non-IRS sites, although we could not rule out a null or negative association in these months.” Reading Figure 5, isn’t it that the upper bound confidence interval for the IRR crossed 1 and the IRR point estimate doesn’t cross 1 until March 2021. It would be helpful to clarify this in the text.

8. Line 323: The reasons for the increase in malaria cases could be reorganized to create a stronger argument. A clearer argument could be restructured as:

a. Clothianidin may not be as effective as pirimiphos-methyl (note Dokolo and only ento studies so far) Could add some information about the clothianidin mechanism and why it may not be as effective – that is slower acting than pirimiphos-methyl.

b. Increasing pyrethroid resistance may have rendered the 2020-2021 UCC campaign less effective than the 2017 campaign, resulting in a loss of the “dual impact” of IRS and LLINs. (cite trends in pyrethroid resistance 2017 to 2020-2021).

The statement on the loss of the combined impact of a dual-intervention with pirimiphos-methyl and pyrethroid LLINs isn’t very strong. There is still a combined intervention with a clothianidin product and pyrethroid LLINs. The issue could be the that pyrethroid LLINs are less effective and no longer contributing to the combined impact. This is why I would recommend removing this point and focusing in on the other two points above.

Then I would consider a separate paragraph to discuss the comparative impact of clothianidin + pyrethroid LLINs vs. next generation LLINs. There is also no data on this comparison and I think this an important area that this paper hints at as well. Could next generation LLINs be a more cost-effective vector control intervention when compared with clothianidin + pyrethroid LLINs? This is a good area for future research.

9. Line 365: Rather than just stating that is unlikely that mosquito behavior changed you could cite the recent PMI VectorLink entomological reports from 2019 and 2020, and perhaps other years, which include data from Lira and Otuke.

10. Line 372: I think these are great references, but a bit confusing to the argument. The near collapse of Anopheles gambiae and Anopheles funestus data is from 2016. It makes one curious as to what these trends look like in the 2020-2021 ‘resurgence’ era compared to previous years. Could this data be gleaned from existing published literature or PMI VectorLink entomological reports over time?

11. Line 391: Entomological measures and residual efficacy data is available from at least some entomological sites within the study areas through PMI VectorLink entomological reports.

12. S1: Should explain in the notes how coverage is defined. Is it structures sprayed out of structures found, or population protected out of total population? There doesn’t seem to be a standard definition in the literature so it would be good to clarify.

13. The authors note that the comparison study objective 2, while focused on IRS, is actually IRS + pyrethroid LLINs vs. next generation LLINs. Do you have documentation on the LLIN coverage in these districts or the relative coverage in the pyrethroid LLIN vs. next generation LLIN districts? The text states that nationally coverage was high, but coverage can vary by area. It would be helpful to note in the manuscript either the coverage rates across the pyrethroid LLIN vs. next generation LLIN districts or that the data was not available at this level.

14. Throughout the paper the terminology changes from “sites” to “health facilities” to “Malaria Reference Centers”. It might be nice to standardize this terminology if possible.

Reviewer #2: • Please clearly state, define, and use the primary outcome measures instead of using vague terms such as “malaria burden”, “burden of malaria” throughout the manuscript, importantly in the abstract.

• It’s essential to add a table or two clearly showing the crude and adjusted IRR for both objectives 1 and 2, including only the lab-confirmed cases. The summary statistics presented in Tables 1 and 2 could be presented as supplementary materials.

• Please show the precision estimates for each point estimate of x% reduction or increase in IRRs.

• Please justify and discuss the choice of your analytical approach: mixed effects negative binomial regression models vs. interrupted time-series ( see example by Ashton et al 2019 below).

• Please take this opportunity to highlight and strengthen the importance of using routine surveillance data in your discussion and citing relevant literature. Here are a few:

o Ashton RA. Use of routine health information system data to evaluate impact of malaria control interventions in Zanzibar, Tanzania from 2000 to 2015. EClinicalMedicine. 2019;12:11–9.

o Ashton R.A. et al. Methodological considerations for use of routine health information system data to evaluate malaria program impact in an era of declining malaria transmission. Am J Trop Med Hyg. 2017; 97: 46-57

o Bennett A. et al. A methodological framework for the improved use of routine health system data to evaluate national malaria control programs: evidence from Zambia. Popul Health Metr. 2014; 12: 30

o Bhattarai A, Ali AS, Kachur SP, Mårtensson A, Abbas AK, Khatib R, et al. (2007) Impact of Artemisinin-Based Combination Therapy and Insecticide-Treated Nets on Malaria Burden in Zanzibar. PLoS Med 4(11): e309.

• Please also elaborate what “high quality data” means and how data quality of assessment was done for this analysis.

• The upper limit of 95% CI for non-IRS sites in year 6 is slightly different in the abstract (0.64) and results (0.63).

6. PLOS authors have the option to publish the peer review history of their article (what does this mean?). If published, this will include your full peer review and any attached files.

**Do you want your identity to be public for this peer review?** For information about this choice, including consent withdrawal, please see our Privacy Policy.

Reviewer #1: No

Reviewer #2: No

---

## [Decision Letter · Decision Letter 1]

28 Jul 2022

Resurgence of malaria in Uganda despite sustained indoor residual spraying and repeated long lasting insecticidal net distributions

PGPH-D-22-00463R1

Dear Dr Epstein,

We are pleased to inform you that your manuscript 'Resurgence of malaria in Uganda despite sustained indoor residual spraying and repeated long lasting insecticidal net distributions' has been provisionally accepted for publication in PLOS Global Public Health.

Best regards,

Ruth Ashton, Ph.D.

Academic Editor

Reviewer Comments (if any, and for reference):

Reviewer's Responses to Questions

**Comments to the Author**

1. If the authors have adequately addressed your comments raised in a previous round of review and you feel that this manuscript is now acceptable for publication, you may indicate that here to bypass the “Comments to the Author” section, enter your conflict of interest statement in the “Confidential to Editor” section, and submit your "Accept" recommendation.

Reviewer #1: All comments have been addressed

Reviewer #2: All comments have been addressed

2. Does this manuscript meet PLOS Global Public Health’s publication criteria? Is the manuscript technically sound, and do the data support the conclusions? The manuscript must describe methodologically and ethically rigorous research with conclusions that are appropriately drawn based on the data presented.

Reviewer #1: Yes

Reviewer #2: Yes

3. Has the statistical analysis been performed appropriately and rigorously?

Reviewer #1: Yes

Reviewer #2: Yes

4. Have the authors made all data underlying the findings in their manuscript fully available (please refer to the Data Availability Statement at the start of the manuscript PDF file)?

Reviewer #1: Yes

Reviewer #2: Yes

5. Is the manuscript presented in an intelligible fashion and written in standard English?

Reviewer #1: Yes

Reviewer #2: Yes

6. Review Comments to the Author

Reviewer #1: No additional comments. All comments have been sufficiently addressed.

Reviewer #2: none.

7. PLOS authors have the option to publish the peer review history of their article (what does this mean?). If published, this will include your full peer review and any attached files.

**Do you want your identity to be public for this peer review?** For information about this choice, including consent withdrawal, please see our Privacy Policy.

Reviewer #1: No

Reviewer #2: No
